# Electrospun PHB/Chitosan Composite Fibrous Membrane and Its Degradation Behaviours in Different pH Conditions

**DOI:** 10.3390/jfb13020058

**Published:** 2022-05-13

**Authors:** Yansheng Zhou, Ying Li, Daqing Li, Yidan Yin, Fenglei Zhou

**Affiliations:** 1Institute for Materials Discovery, Faculty of Mathematical Physical Sciences, University College London, 107 Roberts Building, Malet Place, London WC1E 7JE, UK; 2Spinal Repair Unit, Department of Brain Repair and Rehabilitation, UCL Institute of Neurology, Queen Square, London WC1N 3BG, UK; ying.li@ucl.ac.uk (Y.L.); daqing.li@ucl.ac.uk (D.L.); 3Department of Chemistry, Christopher Ingold Building, University College London (UCL), 20 Gordon Street, London WC1H 0AJ, UK; yidan.yin@ucl.ac.uk; 4Centre for Medical Image Computing, University College London, London WC1V 6LJ, UK

**Keywords:** electrospinning, Polyhydroxybutyrate (PHB), chitosan (CS), pH value, degradation

## Abstract

Peripheral nerve injury (PNI) is a neurological disorder that causes more than 9 million patients to suffer from dysfunction of moving and sensing. Using biodegradable polymers to fabricate an artificial nerve conduit that replicates the environment of the extracellular matrix and guides neuron regeneration through the damaged sites has been researched for decades and has led to promising but primarily pre-clinical outcomes. However, few peripheral nerve conduits (PNCs) have been constructed from controllable biodegradable polymeric materials that can maintain their structural integrity or completely degrade during and after nerve regeneration respectively. In this work, a novel PNC candidate material was developed via the electrospinning of polyhydroxy butyrate/chitosan (PHB/CS) composite polymers. An SEM characterisation revealed the resultant PHB/CS nanofibres with 0, 1 and 2 wt/v% CS had less and smaller beads than the nanofibres at 3 wt/v% CS. The water contact angle (WCA) measurement demonstrated that the wettability of PHB/CS electrospun fibres was significantly improved by additional CS. Furthermore, both the thermogravimetric analysis (TGA) and differentiation scanning calorimetry (DSC) results showed that PHB/CS polymers can be blended in a single phase with a trifluoracetic solvent in all compositions. Besides, the reduction in the degradation temperature (from 286.9 to 229.9 °C) and crystallinity (from 81.0% to 52.1%) with increasing contents of CS were further proven. Moreover, we found that the degradability of the PHB/CS nanofibres subjected to different pH values rated in the order of acidic > alkaline > phosphate buffer solution (PBS). Based on these findings, it can be concluded that PHB/CS electrospun fibres with variable blending ratios may be used for designing PNCs with controlled biodegradability.

## 1. Introduction

Electrospinning is a versatile technique that can efficiently fabricate micro and nanosized polymer fibres. A nanofibrous scaffold can effectively mimic the microstructure in the extracellular matrix based on its structural properties. Electrospun fibres have unique microstructure, including a high surface-to-volume ratio and porosity, therefore, can be used in electronic devices, filters, tissue implants, drug delivery and other biomaterial applications [1,2,3]. Electrospun-fibre-based biomaterials have been extensively researched in the biomedical field, especially in tissue engineering. In nerve tissue engineering, nanosized fibres can provide nerve cell adhesion sites, guide and extend nerve cells from the proximal to distal ends at injury sites. 

Polyhydroxybutyrate (PHB) has been reported as a novel biodegradable polymer that is used in biomedical applications, such as biocontrol agents, drug carriers, tissue implants, anti-osteoporosis agents, etc. [4,5,6]. PHB has satisfying mechanical and biological properties, such as high tensile and compressive strength, stiffness, good biocompatibility and bioactivity [7]. PHB has attracted the attention of implant materials in tissue regeneration, such as nerves and bone [3,4]. Young et al. fabricated PHB-based nerve conduits for repairing long-nerve gap peripheral nerve injuries. The results indicated that PHB conduits can support 4 cm axon regeneration in a nerve gap 63 days after implantation [8]. Mohanna et al. combined PHB with glial growth factors (GGF) to fabricate 2 to 4 cm nerve conduits; the results showed that GGF-containing conduits promoted sustained axonal regeneration and improved target muscle reinnervation. However, due to its high degree of crystallinity, the hydrophobic surface of PHB inhibited the extension growth of cells and attracting of growth factors [8,9]. 

Chitosan (CS) has gradually attracted enormous interest in tissue engineering and regeneration over the last few years because it shows several advantages, including biocompatibility, biodegradability, hydrophilicity, nontoxicity and anti-microbial activity with cells [10]. However, the mechanically inferior feature of CS has limited its usage. Therefore, it is reasonable that both PHB and CS showed mutual complementary potentials based on their properties [11,12].

Both PHB and PHB/CS electrospun fibres have been reported to be used as nerve conduits with growth factors that successfully guided neurite outgrowth after two months of implantation [8,9]. However, due to their hydrophobic surface properties and high crystallinity, they can undergo low hydrolysis degradation and cause a long-term effect after being implanted. Moreover, the hydrophobic surface can limit the ability of cell adhesion and the encapsulation of growth factors without a post-surface treatment. CS was also reported as a porous scaffold to promote neuron proliferation and differentiation in vitro [13], but its fast degradation rate can result in structural collapse and limitation in vivo implantations. A combination of these two materials with different chemical properties in formed electrospun fibres can potentially improve its surface hydrophilicity and adjust in vitro degradability.

Many studies have been carried out on either the hydrolytic or enzymatic degradation behaviours of biodegradable polymers such as polylactic acid (PLA), polycaprolactone (PCL), poly(lactic-co-glycolic acid) (PLGA) and others [14,15,16]. For PHB, like most biodegradable polymers, several factors, such as the degree of crystallinity, hydrophilicity, molecular weight, temperature, and pH of surrounding media, etc., can strongly affect the time that PHB requires to complete degradation. However, previous studies on PHB degradation have been very limited. Wagner et al. illustrated the thermal degradation of PHB from different industrial processes [17]. Chen et al. combined hydroxyapatite (HA) or tricalcium phosphate (TCP) with PHB–poly-hydroxyvalerate (PHB–PHV) and analysed its thermal degradation [18].

In previous studies, a PHB/CS blend solution was optimized and fabricated by electrospinning into nanofibres for tissue engineering applications [12,13,19,20,21]. The effect of the concentration of CS on the microstructure and surface wettability of resultant electrospun PHB/CS nanofibres was analysed [21]. The results demonstrated that increasing CS can enlarge the nanofibres sizes of PHB and reduce the hydrophobic properties. However, its degradation behaviours were still not understood. Therefore, in this report, we investigate the in vitro degradation behaviours of PHB/CS composite nanofibrous membranes with different CS contents under three conditions: neutral (pure pH 7.2 phosphate buffer solution (PBS)), acidic (pH 2 sodium lactate + HCl solution) and alkaline (pH 10 NaOH solution).

## 2. Materials and Methods

Materials

Polyhydroxybutyrate (PHB) (Mw ≈ 660,000 g/mol, Merck), trifluoroacetic acid (TFA) (Reagent Plus^®^, 99%, Merck), chitosan (CS) (medium Mw, Sigma-Aldrich, St. Louis, MO, USA).

Preparation of PHB/CS solutions

The polymer solution was prepared by dissolving 1.5 g of PHB and 0/0.1/0.2/0.3 g of chitosan separately in 10 mL of TFA and preparing 15 wt% PHB with 0/1/2/3 wt% chitosan in different compositions (Table 1). During the dissolving, the PHB solution was first stirred for 4 h at room temperature at 400 rpm. Then, CS powder was gradually added into the PHB solutions and stirred at 100 rpm overnight until homogenous solutions formed.

Fabrication of PHB/CS nanofibrous mats

Random aligned microfibres were fabricated via single-jet electrospinning, as shown in Figure 1. The solutions were placed in a 5 mL thermo syringe with an 18-gauge needle. During the electrospinning process, the flow rate of all solutions was fixed at 0.5 mL/h. A DC high voltage of 11 kV was applied. The collecting distance was fixed to be 13 cm for all groups.

Degradation analysis in acidic (pH = 2), neutral (pH = 7) and alkaline solutions (pH = 12)

The degradation properties of the electrospun PHB/CS nanofibre samples were analysed in three different solutions: (1) The acidic solution was prepared using acidic lactate sodium (ALS) and adjusting the pH to 2 by adding HCl. (2) The neutral solution was prepared using PBS at pH 7.4. (3) The alkaline solution was prepared by adjusting the pH value of PBS to 10 by adding NaOH. Samples were cut into squares (1 × 1 mm^2^) and pre-dried in a vacuum oven for 24 h before the experiments. The scaffolds were then submerged in 10 mL of the acidic, PBS and alkaline solutions in individual tubes with the screw caps tightened and maintained at 37 °C for 8 weeks. All solutions were changed every week. For each data point, three replicates were prepared to minimize the effect of random errors. The scaffolds were removed at specified intervals for analysis, at which time they were rinsed thoroughly with de-ionized water, dried and placed in an oven at 35 °C for 24 h until a constant weight was obtained.

Scanning electron microscope (SEM)

Different microfibres were analysed using a Hitachi 8230 field emission scanning electron microscope (Tokyo, Japan) after coating the samples with Au in a sputter. The diameters of the fibres and beads were measured from the SEM images using ImageJ software. Fifty measurements were carried out for each sample.

Fourier transform infrared spectroscopy (FTIR)

FTIR spectroscopy was performed to investigate the functional groups present in the electrospun PHB/CS nanofibres. Before the test, PHB/CS electrospun fibres were rinsed with de-ionised water and dried in a vacuum dryer until all the water was removed. The FTIR spectra of the PHB/CS electrospun fibres were obtained from a 5 min scan ranging from 4000 cm^−1^ to 400 cm^−1^ wavenumbers with a 2.0 cm^−1^ resolution.

Contact angle measurements

The wettability of the electrospun fibres mats was evaluated through the sensile drop water contact angle test using an OCA 15 plus contact angle measurement system (Data physics, Filderstadt, Germany) equipped with a CCD camera (precision ± 0.2°). All the measurements were performed by applying ultra-pure water at room temperature and 10–15% humidity. The average values were evaluated based on five random points of each specimen.

Thermalgravimetric analysis (TGA)

The samples were subjected to TGA in a PerkinElmer 2000 instrument under a continuous nitrogen flow of 40 mL/min. Approximately 10 mg samples were prepared and then heated in a ceramic crucible from 50–400 °C at the heating rate of 10 °C/min. 

Differential scanning calorimetry (DSC)

DSC was performed by adding an approximately 2 mg sample to an aluminium crucible. The sample was heated from room temperature (25 °C) to 550 °C at the heating rate of 10 °C/min. Nitrogen was used with a gas flow rate of 20 mL/min. All samples were tested under the same conditions to obtain comparable TGA curves.

Statistical analysis

The significance differences of the degradation results in acidic, neutral and alkaline solutions from PHB/CS electrospun fibres were analysed using a one-way ANOVA test in Prism 8. A *p*-value of less than 0.05 was considered as statistically different.

## 3. Results

### 3.1. Morphology and Size of PHB and PHB/CS

All PHB/CS scaffolds were formed in one single phase and prepared by electrospinning techniques. Figure 2 shows the SEM images of fibre morphology for each composition. In Figure 2a, sample PHB/CS0 showed uniform fibre interconnectivity and diameter (331 ± 14 nm). In Figure 2b,c, when the CS content was increased up to 2 wt/v%, the fibre diameter increased from 331 nm to 692 nm, with more roughness and a larger number of non-uniform fibres and beads formed. However, when CS content reached 3 wt/v% (Figure 2d), the fibre diameter decreased to 401 nm, forming more beaded fibre structures with greater sizes. Furthermore, we found each group presented the beaded structure, ranging from a diameter of 314 nm to 1324 nm with the increasing CS contents from 0 to 3 wt/v% (Table 2).

### 3.2. FTIR

Figure 3 shows the FTIR spectrum of PHB, CS and PHB/CS with different weight ratios. The characteristic of the PHB spectrum band is between 1720–1722 cm^−1^, which was assigned to the stretched vibration of the C=O group. Furthermore, the crystallinity phase of PHB in the FTIR spectrum was shown with the peaks observed at 816–826 cm^−1^, 1276–1278 cm^−1^ and 1720–1722 cm^−1^. The peaks of the amorphous phase arose at 1130 cm^−1^. Interestingly, the symmetric wagging of the CH_3_ group in PHB showed at 1378 cm^−1^. For CS, the characteristic peak was at 3300–3550 cm^−1^, representing the stretching OH and NH groups. The amide I that presented in CS was at 1540 cm^−1^. From Figure 3A,D, the addition of CS caused the C=O peak in pure PHB at 1720 cm^−1^ to shift to a lower wavenumber. Moreover, the peaks at 973 cm^−1^ and 1293 cm^−1^ shifted to higher wavenumbers. After acidic and alkaline degradation, all samples showed a broader peak at 3550 cm^−1^, and a strong stretching C=O signal peak formed at 1750 cm^−1^. This is because the crystalline phase of PHB was disturbed by addition CS, which led to rise in the peak of the amorphous phases. The presented hydrogen bond dominated the increase in the amorphous phases.

### 3.3. WCA Measurements

To evaluate the surface hydrophilicity of the PHB/CS nanofibrous mats with different CS contents, the water contact angles (WCA) on the PHB/were measured consecutively as shown in Figure 4. PHB/CS0 showed the highest contact angles at 89°. With the increasing CS contents, contact angles reduced to 65°.

### 3.4. TGA and DSC Analysis

The TGA curves of all PHB/CS fibrous mats were compared with pure PHB and CS to analyse their variation in degradation temperature. The TGA and DTA curves are shown in Figure 5A,B, and the degradation temperatures of each group are summarized in Table 3. In both PHBCS0 and pure chitosan electrospun fibres, degradation occurred at 286.8 and 287.8 °C, respectively. Besides, PHBCS0 had a second degradation temperature at 349.9 °C. Compared to other literatures, we found our results on PHBCS0 and chitosan reliable [8,12,18,21]. Moreover, PHB/CS blended electrospun fibres were all shifted to lower temperatures at 277 °C, 231 °C and 229 °C with increasing of CS contents from 1 to 3 wt%. However, with the addition of CS, no second degradation temperature was detected. Furthermore, the DSC analysis in Figure 6 shows two endothermic peaks presented in all groups, which represented the first and second crystallisation temperatures. Moreover, Table 4 presents the melting point, enthalpy fusion and crystallinity within different groups. The melting point of PHBCS0 was 268.3 °C with the crystallinity at 81%. These results are similar to the values in the literature [8,11,21]. Table 4 shows the decrease in enthalpy fusion from 125 to 76 J/g. Moreover, a reduction in the degree of crystallinity occurred when chitosan contents increased from 0 to 3 wt/v%.

### 3.5. Degradation in Acidic, PBS and Alkaline 

The in vitro characterisation of degradation with various pH levels illustrated in Figure 7. In PBS degradation, the remaining mass percentages at week 12 in each sample were 73.12 ± 2.63 (PHBCS0), 66.32 ± 4.48 (PHBCS1), 65.26 ± 2.51 (PHBCS2) and 43.79 ± 5.61 (PHBCS3). In week 12, the results showed statistically greater degradation of PHBCS3 than PHBCS0, but no statistically significant difference was detected between PHBCS1, PHBCS2 and PHBCS0. In acidic conditions (pH = 2), the percentages of mass remaining at week 12 were 42.57 ± 4.85 (PHBCS0), 31.28 ± 8.85 (PHBCS1), 4.145 ± 7.18 (PHBCS2) and 0 (PHBCS3). In week 12, compared to PHBCS0, there were significant differences in the remaining mass of PHBCS3 (*p* < 0.0001) and PHBCS2 (*p* < 0.0005), but no critical difference was detected in PHBCS1 (*p* > 0.01). In the alkaline conditions, the remaining mass percentages at week 12 in each sample were 71.33 ± 1.14 (PHBCS0), 19.90 ± 2.3 (PHBCS1), 13.28 ± 5.48 (PHBCS2) and 0 (PHBCS3). The results showed significant differences in the remaining mass in all other samples compared to PHBCS0 (*p* < 0.0001).

### 3.6. SEM of PHB/CS under 12 Weeks of Acidic, PBS and Alkaline Degradation

Figure 8 and Table 5 show the SEM images of the PHB/CS electrospun nanofibres and their fibre diameters in PBS, acidic and alkaline solutions after 12 weeks of degradation. In both acidic and alkaline degradations, nano-fibrous structures were gradually reduced with the increasing CS contents. Moreover, in acidic conditions, the PHB/CS electrospun scaffold completely lost its nanofibrous structure in all samples, whereas in the alkaline and PBS conditions, a clear nanofibrous structure could been observed in PHBCS0, PHBCS1 and PHBCS2.

## 4. Discussion

In FTIR, an O-H stretching bond at 3550 cm^−1^ appeared in alkaline solution after 4 weeks of degradation. This is because NaOH can activate the hydroxide groups on PHB/CS fibres’ surfaces [22,23]. Moreover, the signals of the C=O peaks in all samples had a great enhancement because of the polyester degradation in PHB. Furthermore, from the appendix of PHB, the peak at 1227 cm^−1^ was assigned to the absorption of helical (α type) crystals, the 979 cm^−1^ peak was also assigned to the crystalline phase at C-C stretching, and the peak at 1180 cm^−1^ was assigned to the amorphous phase. In our results, the peaks at 973 cm^−1^ and 1293 cm^−1^ shifted to higher wavenumbers because the crystalline phase of PHB was disturbed by the addition of CS, which led to the peak of the amorphous phases increasing. The presented intra-molecular hydrogen bond dominated decrease in the amorphous phases [24].

In Figure 6, a single degradation’s temperature was shown in the TGA/DTA plots of all PHB/CS composites after blending and shifted to the right once the content of CS increased. This is because the proper miscibility occurred between PHB and CS at different compositions, and the addition of CS disrupted the regularity of PHB and reduced the crystallinity [8,11,13,21]. This can be confirmed by the reduction in H_m_ in the DSC curves. Furthermore, the reduction in crystallinity from PHBCS0 to PHBCS3 can be explained by the fact that the rigid CS molecules disrupted the lamellar thickness of the PHB crystallites and suppressed PHB crystallisation [25,26].

In the WCA analysis, the increased CS contents demonstrated a higher degree of hydrophilicity, which is based on the increasing number of terminal hydroxyl and amino groups of the CS chains, which can easily interact with water molecules via H-bonding.

For the in vitro degradation, PHB/CS showed different behaviours with different CS contents under different pH levels. With the increased CS contents, both PHB and CS indicated an increase in the degradation rate at different pH levels. This may be explained that the loss of CS increased the water diffusion possibility, which increased the contact areas between PHB and water. Moreover, a reduction in crystallinity and an increase in hydrophilicity can accelerate the hydrolysis in PHB/CS scaffolds. Compared to PBS (pH = 7) solution, PHB/CS degraded much faster in both sodium lactate/HCl (pH = 2) and NaOH solutions (pH = 10). Interestingly, PHBCS3 completely broke into pieces after 12 weeks of degradation in both acidic and alkaline solutions. This is because the reduction in the crystallinity and hydrophobic nature of PHBCS3 which resulted in faster water penetration through the structure. Besides, in acidic conditions, samples with additional CS experienced a much faster degradation rate than those without CS. This is due to the weak base nature of CS [27,28] that dissolved in acidic solutions and led to decrease in CS contents in the scaffold and more water penetration. In the base solution, PHBCS0 was broken down small pieces after 12 weeks of degradation, whereas PHBCS1, PHBCS2 and PHBCS3 retained their structure integrity. This is because pure PHB was degraded into carboxylic acid and alcohol in the base solution, and hydroxyl anions can remove protons from the acid and produce negatively charged carboxylate ions [27]. Ian et al. also reported similar trends for highly concentrated alkaline solutions (0.1–4 M NaOH). The hydroxyl anions in the solution lowered the activation energy of the ester bonds. CS, with cation groups, attracted OH- groups in the solution and lowered the alkali pH values [28].

## 5. Conclusions

This research has discovered the physical, chemical and microstructure behaviours of PHB/CS electrospun fibres under acidic, neutral, and alkaline conditions with different CS contents, which provides a view of the nano-surface morphology change of PHB/CS. We found that the PHB/CS electrospun mats were successfully fabricated by the electrospinning process as a single phase in each composition. Moreover, with increasing additions of CS, PHB/CS electrospun fibres have reduction in both hydrophobic nature and crystallinity. Furthermore, we analysed the degradability of PHB/CS in acidic, neutral, and alkaline conditions. The overall trends showed that the degradation rate increased with the addition of CS at all pH values. Furthermore, the degradability of PHB/CS between different pH values were ranked from acidic > alkaline > phosphate buffer solution (PBS). Both acidic and alkaline environments will accelerate PHB/CS degradation, and completely dissolve in pieces of at week 12. Comparatively, acidic environments can perform a more violent degradation of the PHBCS scaffold than in alkaline and PBS solutions. 

## Figures and Tables

**Figure 1 jfb-13-00058-f001:**
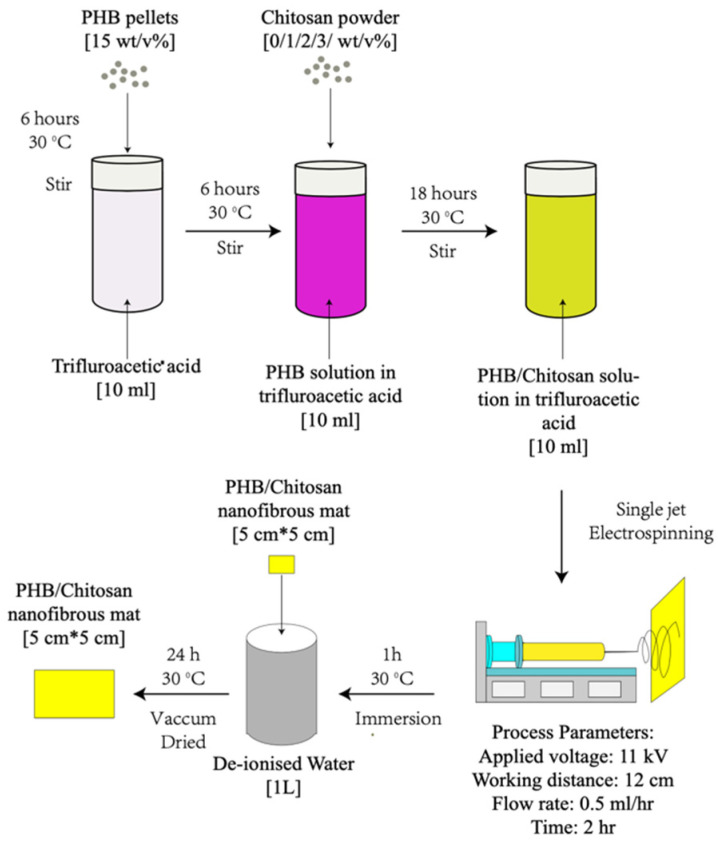
Working flow chart of preparing PHB/CS solution.

**Figure 2 jfb-13-00058-f002:**
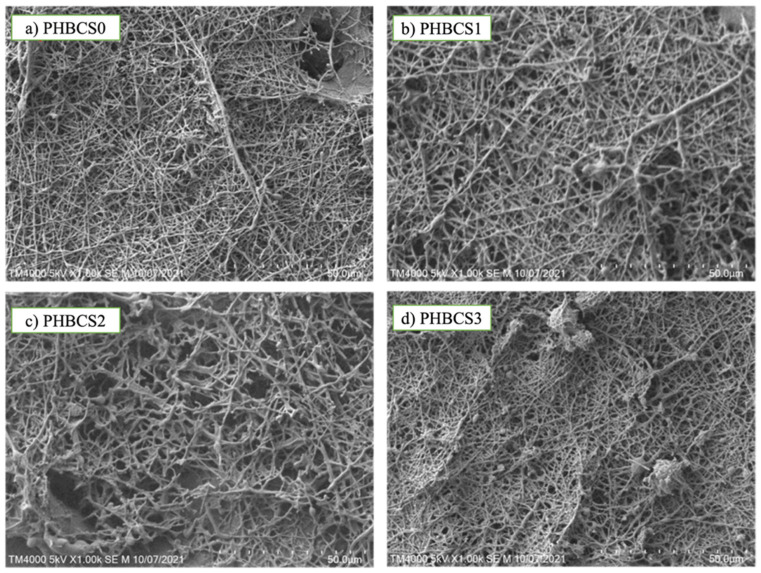
SEM images of PHB/CS electrospun scaffolds with different CS contents (**a**) 0 wt/v%, (**b**) 1 wt/v%, (**c**) 2 wt/v% and (**d**) 3 wt/v%.

**Figure 3 jfb-13-00058-f003:**
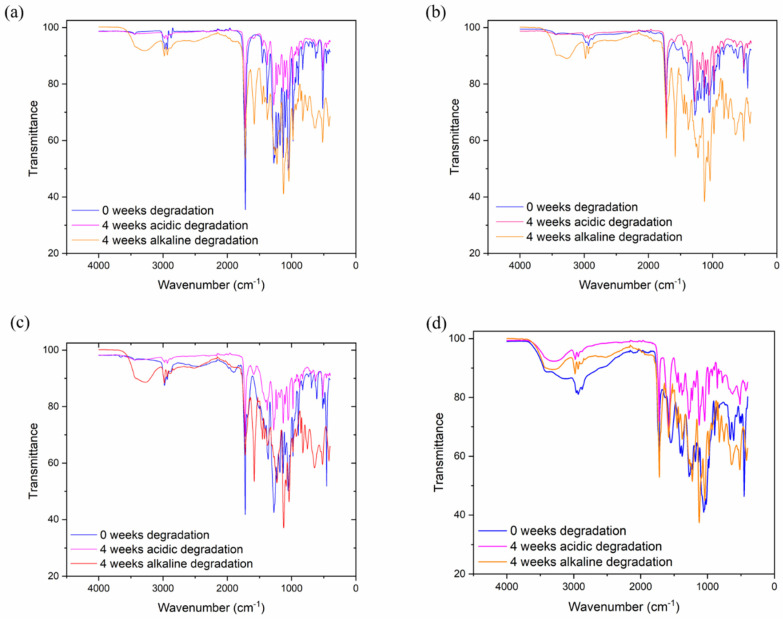
FTIR spectra of PHB/CS electrospun scaffolds. (**a**) PHBCS0, (**b**) PHBCS1, (**b**) PHBCS2, (**d**) PHBCS3.

**Figure 4 jfb-13-00058-f004:**
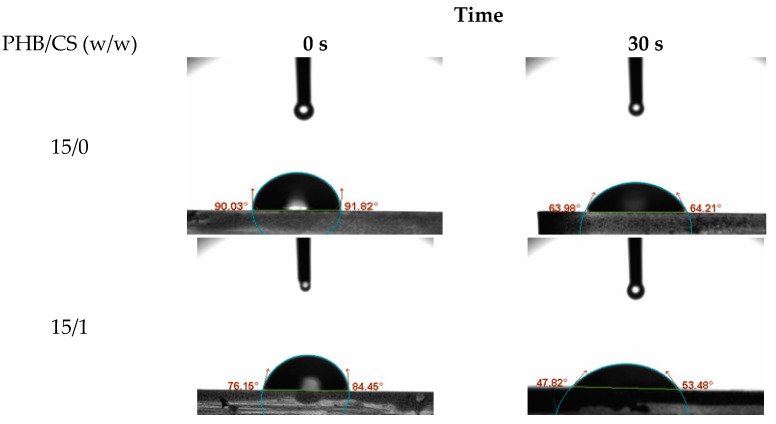
Water contact angle measurements.

**Figure 5 jfb-13-00058-f005:**
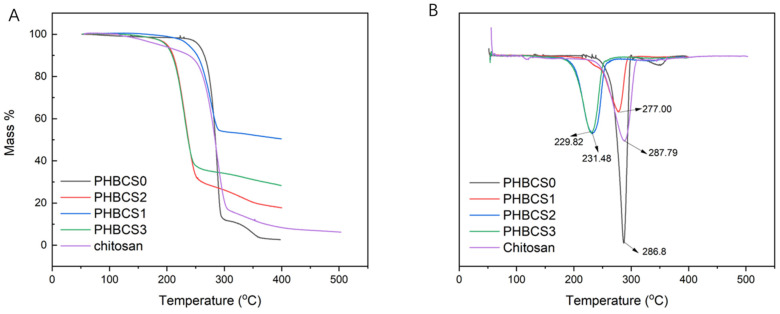
(**A**) TGA and DTA graphs of PHB/CS composite films with different CS contents. (**B**) DSC graphs of PHB/CS composite films with different CS contents.

**Figure 6 jfb-13-00058-f006:**
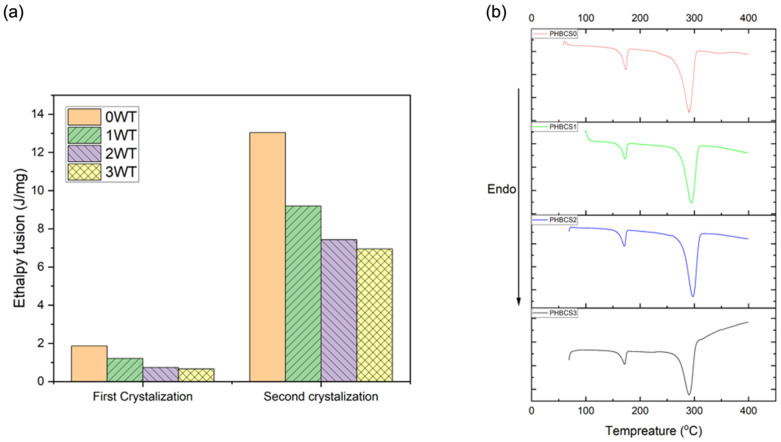
(**a**) Enthalpy fusion and (**b**) DSC curves of PHB/CS samples.

**Figure 7 jfb-13-00058-f007:**
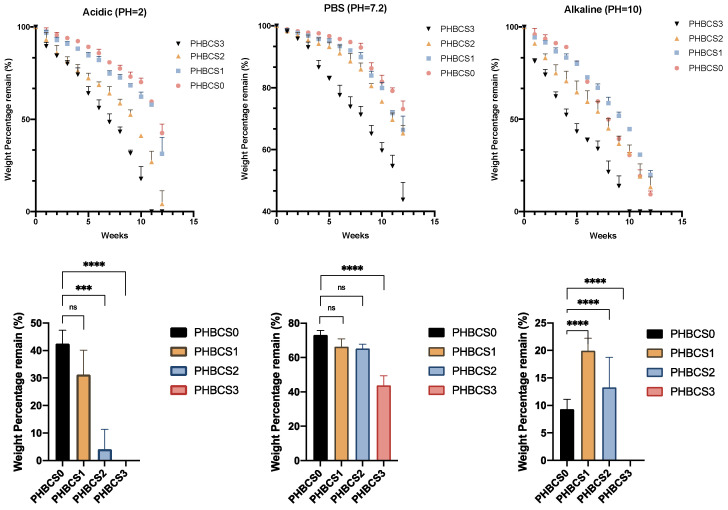
Remaining weight of PHB and PHB/CS electrospun mats between different pH levels in week 12. (**** (*p* < 0.0001), *** (*p* < 0.0005), ns (*p* > 0.1)).

**Figure 8 jfb-13-00058-f008:**
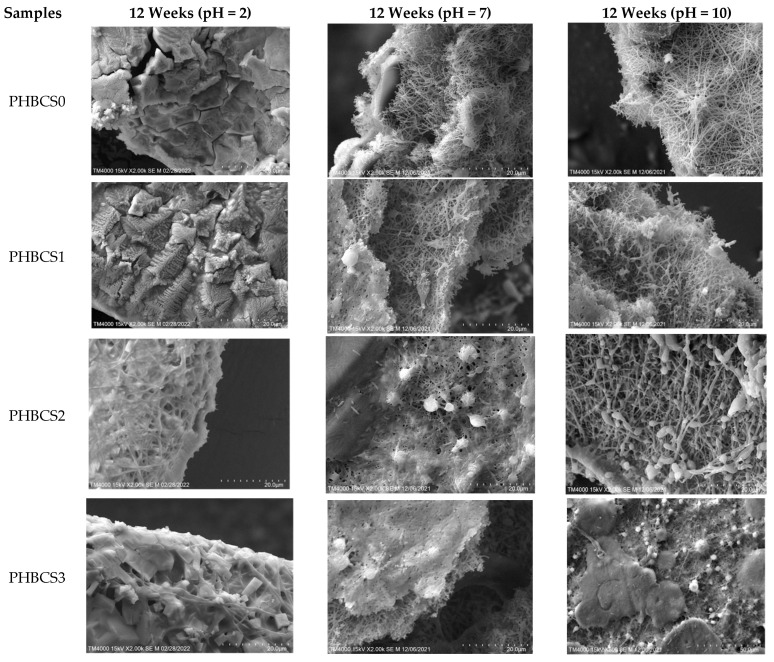
Remaining weight of PHB and PHB/CS electrospun mats in week 12.

**Table 1 jfb-13-00058-t001:** Concentration of PHB and CS in different groups.

Groups	Conc. of PHB (wt/v%)	Conc. of CS (wt/v%)
PHB/CS0	15	0
PHB/CS1	15	1
PHB/CS2	15	2
PHB/CS3	15	3

**Table 2 jfb-13-00058-t002:** Average fibre and bead diameter of PHB/CS nanofibrous membranes with different CS contents.

Groups	Average Fibre Diameter (nm)	Bead Diameters (nm)
PHB/CS0	331 ± 14	314 ± 25
PHB/CS1	428 ± 23	520 ± 63
PHB/CS2	692 ± 101	538 ± 24
PHB/CS3	401 ± 58	1324 ± 58

**Table 3 jfb-13-00058-t003:** Degradation temperature of PHB/CS samples.

Groups	First Degradation Temperature (°C)	Second Degradation Temperature (°C)
PHBCS0	286.8	349.9
PHBCS1	277.0	/
PHBCS2	231.5	/
PHBCS3	229.9	/
CS	287.8	/

**Table 4 jfb-13-00058-t004:** DSC graphs of PHB/CS composite films with different CS contents.

Groups	Tm (°C)	Hm (J/g)	Crystallinity (%)
PHBCS0	268.3	125.0	81.0
PHBCS1	272.6	104.1	71.3
PHBCS2	276.4	81.60	55.9
PHBCS3	279.7	76.06	52.1

**Table 5 jfb-13-00058-t005:** Average fibre diameter of PHB/CS nanofibrous membranes with different CS contents after 12 weeks of degradation in different pH values.

pH	Groups	Average Fibre Diameter (nm)
2	PHBCS0	/
	PHBCS1	/
	PHBCS2	321 ± 21
	PHBCS3	502 ± 103
7	PHBCS0	241 ± 13
	PHBCS1	401 ± 29
	PHBCS2	391 ± 34
	PHBCS3	388 ± 39
10	PHBCS0	298 ± 13
	PHBCS1	309 ± 89
	PHBCS2	545 ± 44
	PHBCS3	310 ± 98

## Data Availability

The data presented in this study are available on request from the corresponding author. The data are not publicly available due to the restriction of privacy terms in Institute of Material Discovery.

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
