# Peer review of "Electrospun PHB/Chitosan Composite Fibrous Membrane and Its Degradation Behaviours in Different pH Conditions"

_jfb, 2022, doi:10.3390/jfb13020058_

Round 1

Reviewer 1 Report

The paper entitled "Properties of a biodegradable mixture of co-electrophized fibers of PHB and Chitosan and its degradation behavior under acidic and alkaline physiological conditions" presents studies of nanofibers produced with PHB and chitosan, with the purpose of which this blend can improve the physical, chemical and mechanical characteristics of the material.

In general, the paper is confusing.

There are several formatting issues (line 88, 165, 166, 208).

There is no discussion of the results obtained, no type of comparison with other literature, even, there are only 7 bibliographic references in the entire text, which is more than 15 years old.

For these reasons, I do not recommend the publication of this paper.

Author Response

Responses to Reviewer 1:

Thanks very much for taking your time to review this manuscript and sorry for the confusion caused. We really appreciate all your comments and suggestions. Please find my detailed responses below. All revisions/corrections have been made in the re-submitted manuscript.

  1. the study reported in this manuscript aims to develop PHB/chitosan composite nanofibres with proper degradability as a novel scaffold candidate for PNS repair. This point has been made clear in the Introduction section.
  2. we for the first time investigated how the chitosan content affected PHB/CS electrospun fibers’ degradation behaviors under pH = 2, 7 and 10, (i.e. acidic, neutral, and alkaline conditions). In this study, we characterized the electrospun PHB/Chitosan fibers by using SEM to assess its surface morphology changes of samples under different conditions; we also employed FTIR spectra to determine the crystallinity variation of electrospun PHB/Chitosan after degradation  under different pH values for 4 weeks;the degradation rates of each composition of four PHB/Chitosan composite nanofibres were determined and ploted against the pH values.

3 we believe this work reported in this manuscript is worthy to publish in the Journal of Functional Biomaterials because

1) As shown in Figure 10, after 12 weeks degradation, nanofibrous structure of PHB/CS electrospun mats has gradually disappeared and formed in a solid film. This causes a reduction of surface-to-volume ratio which will reduce the implanted cell adhesion and migration. Also, lack of porosity and surface roughness of all samples after 12 weeks degradation has shown in figure 10. these will make cell hard to adhere and infiltration through the scaffolds.   

2) In physiological conditions, inflammatory factors at injured sites can increase the lactic acid secretion and lower the pH values. Therefore, acidic degradation will not only affect the structural integrity and surface morphology of electrospun fibers, but also accelerate the degradation.

3) There is few research has mentioned about the degradation behaviors of electrospun fibers in different pH conditions.

4) In line 88 (now line 82) , the ‘table 1’ has been corrected.

5) In line 165 and 166 (now line 155 and 156), the formatting issues has been corrected

6) in line 208, the format issues have been corrected

Reviewer 2 Report

Dear Authors,

The manuscript entitled "Properties of a biodegradable blended PHB and Chitosan co-electrospun fibers and its degradation behaviors under acidic and alkaline physiological conditions” is an interesting topic. However, considerable changes are needed:

  1. Changing the title, making it shorter, would be more appealing
  2. Table 1: is not correctly positioned
  3. Line 66: PBS is not defined
  4. Line 66: change PH to pH
  5. Line 114 and 118: remove italic formatting
  6. figure 1: It isn’t referring in the text

  1. The article is not in its final version, please send the final version without comments!!

Author Response

Response to Reviewer 2:

Thanks very much for taking your time to review this manuscript. Our detailed responses are given below:

  1. For the title, we changed from ‘Properties of a biodegradable mixture of co-electrospun fibers of PHB and Chitosan and its degradation behavior under acidic and alkaline physiological conditions" to ‘Electrospun PHB/Chitosan composite fibrous membrane and its degradation behaviors in different pH conditions’
  2. Table1 has been re-positioned in the revised manuscript
  3. For line 66, defined PBS is the abbreviation of phosphate buffer solution
  4. For Line 66 and other places, all ‘PH’ was corrected to ‘pH’.
  5. For Line 114 and 118, we corrected them into formal text form.
  6. For figure 1, it is now referred in the sentence in line 47.

Reviewer 3 Report

In the present manuscript the authors describe the preparation and the degradation in different acqueous media of biodegradable blended poly hydroxybutyrate and Chitosan. The manuscript it well-written and flows well. The experimental data are motivated, well-reported and justified. Nevertheless, the introduction needs to be more robust on the side of the properties of PHB and Chitosan in biomedical applications.  This would also enlarge the number of references to the manuscript which, currently is limited to 7. Therefore, I would recommend the authors to review the above-mentioned points along with the below listed minor points before recommending publication.

Minor points to be addressed:

  • PH is written along the text instead of pH
  • a space is needed between numbers and units
  • cm-1 is written along the text instead of cm-1
  • page 2 line 57 “Polyhydroxybutryate” should be revised as “Polyhydroxybutyrate”
  • page 2 line 66 remove ‘in”
  • page 2 line 70 “inflammatory” should be revised as “inflammation”
  • page 2 line 88 “table1?” should be revised as “table 1”
  • page 3 line 107 “will be changed every week. For each data point, three replicates will” should be revised as “were changed every week. For each data point, there replicates were”
  • page 4 line 197 “In figure 1b and c” should be revised as “In figures 1b and 1c”
  • page 4 line 213 “ CH3” should be revised as “CH3

Author Response

Response to Reviewer 3:

Thanks very much for taking your time to review this manuscript.  our detailed responses are given below:

  1. We have corrected the punctuation.
  2. we have made the clarification in abstract, with alkaline>acidic>phosphate buffer solution (PBS).
  3. All the language issues have been checked and corrected
  4. All the typos have been corrected.
  5. We have changed the inconsistent styles including bold fonts and spaces.
  6. The units in all paper and table 2 (now is table 3) are all corrected.

we have included more relevant references (in total 24) including some papers published in 2022. However, it is worth pointing out that for PHB/chitosan nanofibre scaffoldss, there have very limited  publications focused on the degradation study in various conditions.

Reviewer 4 Report

The manuscript "Properties of a biodegradable blended PHB and Chitosan co-electrospun fibers and its degradation behaviors under acidic and alkaline physiological conditions" reads as if it were written in a rush. The content of the manuscript, however, is of importance and interest. In my opinion, the manuscripts has quite many issues, for example and cannot be published in the current state:
PH should be written as pH;
Inconsistency of punctuation.
alkaline > acidic > phosphate buffer -- marked from most aggressive to least aggresive? Clarify
language issues
Typos
Inconsistent style i.e. bold and spaces
No superscripts in units
Table 2 -- temperature column, not clear
Insufficient literature overview -- only 7 sources, none published in the last  years.
//  Major Revision

Author Response

Response to Reviewer 4

Thanks very much for taking your time to review this manuscript.. Our responses are gives below:

See my comments on the right.

  1. In the introduction parts (from 46 to 71), we have added more recent reviews on the PHB and Chitosan in biomedical application.
  2. we have included more relevant references (in total 24) including some papers published in 2022. However, it is worth pointing out that for PHB/Chitosan nanofibre scaffolds, there have very limited publications focused on the degradation study in various conditions.
  3. All the ‘PH’ has changed to ‘pH.
  4. All spacing between numbers and units are added.
  5. Cm-1 are all corrected to cm-1.
  6. Page 2 line 57 (now is in 33,46 and 86) are all corrected.
  7. Page 2 line 66, ‘in’ has been removed.
  8. page 2 line 70 the inflammatory has been deleted.
  9. page 2 line 88 (now line 82) , the ‘table 1’ has been corrected.
  10. page 3 line 107 (now line 114), the sentenced has been corrected.
  11. Page 4 line 197 (now line 160), the “In figure 1b and c” has revised as “In figures 2b and 2c”.

Page 4 line 213 (now line 183), “ CH3” has been revised as “CH3

Reviewer 5 Report

The manuscript by Zhou et al. reports a study on the preparation of PHB/chitosan (CH) nanofibers prepared by electrospinning method and its characterization. From the materials point of view, the peer-reviewed article is characterized by low scientific novelty, since PHB/CH electrospun fibers were already described in many applications e.g.,  cartilage tissue engineering, wound healing, as well as nervous system replacement (application mentioned by the Authors). Additionally, several works were concerned on the parameters influencing the electrospinning process of PHB/CH blends.

As the title of the publication suggests, the main goal of the work was to analyse the degradation behaviour of the prepared fibres at different pH conditions. Unfortunately, the analysis is very limited and could be much deeper based on generally available work on the degradation of PHB and chitosan e.g., Authors claimed that the “The C=O stretching enhancing is not clearly understood.” (line 235), whereas it is just result of polyester degradation.

Bearing in mind the above arguments, mainly the lack of scientific originality, I am applying for the rejection of the article for publication in Journal of Functional Biomaterials.

Author Response

Response to Reviewer 5:

Thanks very much for taking your time to review this manuscript. We have tried to address all questions as below: The main novelty of this publications is focused on the degradation behaviors of PHB/Chitosan electrospun fibers in different pH levels. This research topic  is important and useful for developing novel PNS scaffolds but has been hardly studied in details in literature published so far. Our study  have demonstrated for the first time that the degradation rate and surface morphology of electrospun PHB/chitosan nanofibres vary with pH conditions.  Especially in surface morphology, the destruction of nanofibrous structure as shown in Figure 10 could significantly affect the tissue regeneration, including cell viability, adhesion, proliferation and differentiation.

In the process parameters, we have tried different, not only process parameters, but also various compositions from different papers to fabricate a uniform, single phase nanofibrous structure. However, because the variant source of PHB, molecular weight and deacetylation of chitosan, it is hardly to repeat the experiment by follow the process parameters and electrospinning from other papers. Therefore, we believe it is necessary to reproduce a new process parameter and composition and analyze.

Besides, in physiological conditions, inflammatory factors at injured sites can increase the lactic acid secretion and lower the pH values. Acidic environment can accelerate the degradation rate of scaffold and destroy the nanofibrous structure after long week implants, and these results has been proven in the research paper. Therefore, it is worth to publish to publish my paper.

Thank for your information and I have rewritten the discussion in FTIR parts from line 263 to 264. In respond to your ‘limited analysis on degradation’, the analysis of degradation rate for PHB/Chitosan samples were mainly focused several key parameters: 1. Surface morphology 2. Composition between PHB and chitosan3. Surface wettability 4. Thermal stability 5. Crystallinity, 6. FTIR analysis and 6. Degradation rate. These have proven that firstly; addition of chitosan could reduce the crystallinity and hydrophobic nature of PHB/CS scaffold and results in faster degradation than normal. Besides, the thermal stability has furtherly proven this result. Secondly, FTIR investigate, not only the variation of function groups between different composition of PHB and chitosan, but also the variation of functional groups before and after 12 weeks degradation in different pH levels. We found the the stretching OH and amide-I groups in 3300-3550 cm-1 could make PHB/Chitosan scaffold have more water penetration. These showed the reasons why degradation is much faster after addition of chitosan and what types of function groups has generated and resulted in further degradation. And thirdly, SEM results has clearly indicated the surface morphology changes between different pH levels. Our results showed that the degradability of PHB/CS between different pH values were ranked from acidic>alkaline>phosphate buffer solution (PBS). Because both acidic, and alkaline environment will accelerate the degradation of PHB/CS and showed a completely dissolved happened in PHBCS3 at week 12. Comparatively, acidic environment can perform more violent degradation on PHBCS scaffold than alkaline and PBS degradation, especially on morphology of nanofibrous structure. Therefore, we believe our results are not limited.

Round 2

Reviewer 1 Report

After the adjustments made to the article, I consider it ready for publication.

Author Response

Thanks for your suggestion and i am happy that you accept this paper.

Reviewer 2 Report

Dear authors,

This version is much clearer and more complete than the previous version. The title “Electrospun PHS/Chitosan composite fibrous membrane and its degradation behaviors in different pH conditions” now presented is clearer and more appealing.

I suggest once again that tables and figures should be referenced in the text before their presentation. Take this care and rectify the whole document.

Best regards

Author Response

Thanks very much for taking your time to review this manuscript. Our responses are gives below:

  1. Figure 1 was referenced in line 122 and 178
  2. Figure 2 was referenced in line 182,185,186,187,190
  3. Figure 3 was referenced in line 208,210,217
  4. Figure 4 was referenced in line 228,230
  5. Figure 5 was referenced in line 234,249
  6. Figure 6 was referenced in line 241,253,301
  7. Figure 7 was referenced in line 258,273
  8. Figure 8 was referenced in line 277,283
  9. Table 1 was referenced in line 101,115
  10. Table 2 was referenced in line 193,195,235
  11. Table 3 was referenced in line 235,251
  12. Table 4 was referenced in line 242,245,255
  13. Table 5 was referenced in line 278,283

Reviewer 3 Report

The authors have addressed my and the other reviewers comments.

Author Response

(The authors gave the same response as above.)

Reviewer 4 Report

The authors have performed a thorough revision of "Properties of a biodegradable blended PHB and Chitosan co-electrospun fibers and its degradation behaviors under acidic and alkaline physiological conditions" after the first round, addressing comments from 5 reviewers. I congratulate the authors on the hard work. In my opinion, the article can be considered for publication.

//Accept

Author Response

(The authors gave the same response as above.)

Reviewer 5 Report

After revision I accept manuscript.

Author Response

(The authors gave the same response as above.)
